# Exploring Few-Shot Image Generation With Minimized Risk of Overfitting

## Abstract

Few-shot image generation (FSIG) using deep generative models (DGMs) presents a significant challenge in accurately estimating the distribution of the target domain with extremely limited samples. Recent work has addressed the problem using a transfer learning approach, *i.e.* fine-tuning, leveraging a DGM that pre-trained on a large-scale source domain dataset, and then adapting it to the target domain with very limited samples. However, despite various proposed regularization techniques, existing frameworks lack a systematic mechanism to analyze the degree of overfitting, relying primarily on empirical validation without rigorous theoretical grounding. We present Few-Shot Diffusion-regularized Representation Learning (FS-DRL), an innovative approach designed to minimize the risk of over-fitting while preserving distribution consistency in target image adaptation. Our method is distinct from conventional methods in two aspects: First, instead of fine-tuning, FS-DRL employs a novel scalable Invariant Guidance Matrix (IGM) during the diffusion process, which acts as a regularizer in the feature space of the model. This IGM is designed to have the same dimensionality as the target images, effectively constraining its capacity and encouraging it to learn a low-dimensional manifold that captures the essential structure of the target domain. Second, our method introduces a controllable parameter called sharing degree, which determines how many target images correspond to each IGM, enabling a fine-grained balance between overfitting risk and model flexibility, thus providing a quantifiable mechanism to analyze and mitigate overfitting. Extensive experiments demonstrate that our approach effectively mitigates overfitting, enabling efficient and robust few-shot learning across diverse domains.

## 1 Introduction

In recent years, Deep Generative Models (DGMs) have achieved remarkable breakthroughs in the generation of high-quality and diverse samples across various domains (Higgins et al., 2016; Karras et al., 2019; Song et al., 2020b; Ruiz et al., 2023). However, reliance on extensive data presents a significant challenge in scenarios where data is scarce (Abdollahzadeh et al., 2023). To address this issue, Few-Shot Image Generation (FSIG) methods (Wang et al., 2018; Zhao et al., 2022) have emerged, aiming to generate diverse images with limited training samples.

Most FSIG methods rely on fine-tuning a DGM, typically a generative adversarial network (GAN) (Goodfellow et al., 2014), which pretrained on a larger and "similar" dataset (Ojha et al., 2021; Zhu et al., 2022; Zhao et al., 2022; 2023). However, this fine-tuning process, which involves adjusting the generator $p_\theta(z)$ to minimize the loss in the target domain $\mathcal{Y}$, $\min_\theta \mathbb{E}_{(z\sim\mathcal{N}(0,I),y\sim\mathcal{Y})}[\mathcal{L}(p_\theta(z),y)]$, often leads to overfitting, visual artifacts, and catastrophic forgetting (Saito et al., 2017; Radford et al., 2015; Kirkpatrick et al., 2017) when only a few samples are available.

More recently, Diffusion Models (DMs) (Ho et al., 2020; Song et al., 2020b) have demonstrated remarkable success, surpassing GANs in image generation (Dhariwal & Nichol, 2021). Their inherent scalability and more stable training process allow DMs to be trained on larger datasets, resulting in superior generalization capabilities. This makes them particularly adept at tasks that require fine-grained detail manipulation, such as text-to-image translation (Saharia et al., 2022; Ramesh et al., 2021) and intricate image editing (Meng et al., 2021). Given these strengths, it is attractive to consider adapting DMs for FSIG, potentially offering superior solutions to existing GAN-dominated methods.

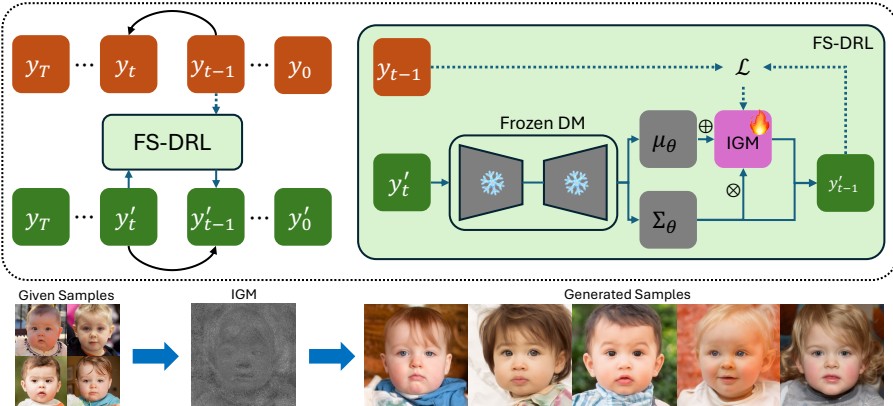

Figure 1: An illustration of FS-DRL, demonstrating how our method overcomes overfitting during IGM training, along with visual showcase. The dotted arrow (top) is used only during training.

However, directly applying current FSIG techniques such as regularization (Li et al., 2020; Ojha et al., 2021) and modulation (Zhao et al., 2022) to DMs proves challenging. The significantly larger number of parameters in DMs and their iterative nature not only fail to address the problems faced by GANs but may exacerbate overfitting and catastrophic forgetting issues (Abdollahzadeh et al., 2023). Consequently, we define the research question as follows: How can we adapt the pre-trained diffusion model to the target domain while minimizing the risk of overfitting?

To address this question, we present Few-Shot Diffusion-Regularized Representation Learning (FS-DRL), as shown in Fig. 1. Our method consists of three main contributions:

**Firstly**, we introduce a novel framework to adapt a pretrained DM to a specific domain. Unlike other approaches that attempt to modify the generator (Wang et al., 2018; Ojha et al., 2021; Zhao et al., 2023), our method is designed to "influence" the generation process. Specifically, given a target domain $\mathcal{Y}$, our method converts the unconditional generation process to a conditional one, and at the diffusion time $t$, the objective is thus $\min_\theta \mathbb{E}_{(q(y_t|y), y \sim \mathcal{Y})} [\mathcal{L}(p_\theta(y_t|\mathcal{Y}), y)]$. We find that introducing a non-adaptive module, which we call the Invariant Gradient Matrix (IGM), is sufficient to achieve our objective by guiding the generation process.

**Secondly**, we theoretically demonstrated that this IGM is essentially equivalent to a "simplest" classifier in classifier-guided diffusion model (Song et al., 2020b). The weights can be seen as an "attention matrix", which determines the amount of "attention" different regions of the state should receive for a specific domain. Furthermore, we introduce a **Scalable** property for IGM, which allows flexible control over granularity. This scalability impacts the trade-off between generalization and specificity. Defining an IGM for multiple images provides high generalization with low overfitting risk, while a single IGM per image offers high specificity but increases overfitting risk.

**Thirdly**, we propose two optimization techniques that significantly enhance the performance of our method in Few-Shot Image Generation (FSIG). The introduction of percentile gradient clipping and simplified loss function allows our approach to achieve comparable results to state-of-the-art methods, with particularly notable improvements in mode coverage. Additionally, we conducted experiments on further parameter reduction, exploring the trade-offs between model complexity and performance.

We summarize the structure of the paper as follows. In Sec. 3.1, we provide a preliminary introduction to diffusion models and formalize the notion we used in this paper. We then introduce the details of our proposed method FS-DRL (Sec. 3.2) with theoretical analysis (Sec. 3.3) and two optimization strategies (Sec. 3.4. In Sec. 4, we demonstrate the effectiveness of our proposed method through empirical comparisons with the baseline, and a comprehensive component analysis.

## 2 RELATED WORKS

**Few-Shot Image Generation** Conventional approaches typically apply fine-tuning a Generative Model pre-trained on a large dataset of a similar domain (Bartunov & Vetrov, 2018; Wang et al., 2018; Clouâtre & Demers, 2019). However, full model fine-tuning typically leads to mode collapse

(Hu et al., 2023). To mitigate this, various selective fine-tuning techniques have been proposed. These include updating only part of the model, *e.g.* freezing the discriminator of GAN (Noguchi & Harada, 2019; Mo et al., 2020), preserving crucial pretrained weights identified by the modulation method and Fisher Information (Li et al., 2020; Zhao et al., 2022; 2023), and maintaining structural similarity between source and target domain distributions (Ojha et al., 2021; Xiao et al., 2022; Hu et al., 2023). GenDA (Mondal et al., 2022) first utilize the representation learning method for FSIG, however, their method is limited to StyleGAN (Karras et al., 2019) as it requires a "short" explicit latent code. CRDI (Cao & Gong, 2024) is the most similar work to ours. However, we showed that their framework can be regarded as a special case of ours with the highest degree of overfitting in Sec. 3.3.

**DM for Representation Learning** There are three main approaches which are close to our proposed method: (1) Diffusion Models with AutoEncoder (VAE) (Kingma & Welling, 2013), this approaches including D2C (Sinha et al., 2021), Diff-AE (Preechakul et al., 2022), DiTi (Yue et al., 2024) *et al.*, which also be able to generate given only a few samples ($\geq 100$), however, these methods require to train a Latent DMs from scratch to adapt a pre-trained VAE, which cause significant computational resources and cannot be applied to varies pre-trained diffusion model. (2) Text-to-Image Diffusion Model (DM), because of the high scalability of DMs, many LMMs such as DALL-E (Ramesh et al., 2021) and Stable Diffusion (Rombach et al., 2022) are also applied to FSIG task. However, existing multimodal foundation models have limited capacity for generating images of unseen categories in inferring. Although methods such as DreamBooth (Ruiz et al., 2023) can generate samples from a few shots, they are limited to adapting at the subject level. (3) Diffusion Inversion, which can be further decomposed into two methods, training-free method including SDEdit (Meng et al., 2021), Edict (Wallace et al., 2023) *et al.* and training-required method including Textual Inversion (Gal et al., 2022), MCG (Chung et al., 2022) *et al.*, these methods are mainly for Image Editing task which requires deterministic inversion, hence not suitable for FSIG as the diversity is a key point.

# 3 METHODOLOGY

## 3.1 PRELIMINARIES

**Diffusion Model** Denoising Diffusion Probabilistic Model (Ho et al., 2020) (DDPM) is a latent variable model that learns to sample from a distribution by learning to iteratively denoise samples. The forward process $q(x_{0:T})$ adds noise to the sample $x_0$ as

$$q\left(\mathbf{x}_t \mid \mathbf{x}_{t-1}\right) = \mathcal{N}\left(\mathbf{x}_t; \sqrt{1-\beta_t}\mathbf{x}_{t-1}, \beta_t\mathbf{I}\right) \tag{1}$$

where $\beta_t$ is pre-defined to control the variance schedule. Song et al. (2020b) and Ho et al. (2020) shown that the reverse process can be converted to a generative model by sampling $x_T \sim \mathcal{N}(\mathbf{0}, \mathbf{I})$ and transforming incrementally into a data manifold as $p_\theta\left(\mathbf{x}_{0:T}\right) = p\left(x_T\right)\prod_{t=1}^{T} p_\theta\left(\mathbf{x}_{t-1} \mid \mathbf{x}_t\right)$, where

$$p_\theta\left(\mathbf{x}_{t-1} \mid \mathbf{x}_t\right) = \mathcal{N}\left(\mathbf{x}_{t-1}; \mu_\theta\left(\mathbf{x}_t, t\right), \Sigma_\theta\left(\mathbf{x}_t, t\right)\right) \tag{2}$$

Here $\mu_\theta$ and $\Sigma_\theta$ are the outputs of a neural network. Furthermore, by using the reparameterization trick and Tweddie's formula (Stein, 1981), we can get two equivalent interpretations

$$\boldsymbol{\mu_\theta}\left(\boldsymbol{x}_t, t\right) = \frac{1}{\sqrt{\alpha_t}}\boldsymbol{x}_t - \frac{1-\alpha_t}{\sqrt{1-\bar{\alpha}_t}\sqrt{\alpha_t}}\boldsymbol{\epsilon_\theta}\left(\boldsymbol{x}_t, t\right) = \frac{1}{\sqrt{\alpha_t}}\boldsymbol{x}_t + \frac{1-\alpha_t}{\sqrt{\alpha_t}}\boldsymbol{s_\theta}\left(\boldsymbol{x}_t, t\right) \tag{3}$$

where $\alpha_t$ is mean coefficient defined as $1 - \beta_t$, $\boldsymbol{\epsilon_\theta}\left(\boldsymbol{x}_t, t\right)$ and $\boldsymbol{s_\theta}\left(\boldsymbol{x}_t, t\right)$ are noise network and score network, respectively. See Luo (2022) and Song et al. (2020b) for complete deviation.

## 3.2 FEW-SHOT DIFFUSION-REGULARIZED REPRESENTATION

**Definition 1.** *(Target Domain Adaptation) Given a diffusion model trained on a source domain dataset $\mathcal{X}$, we say that the diffusion model is adapted to target domain $\mathcal{Y}$ with degree $\eta$ at $t$ when $\mathbb{E}_{x_0 \in \mathcal{X}, y_0 \in \mathcal{Y}}\left[\mathcal{M}(x_0, y_0, t)\right] \geq \eta$, where domain adaptation measure $\mathcal{M}(x_0, y_0, t)$ is defined as:*

$$\mathcal{M}(x_0, y_0, t) := \frac{1}{2}\left(\underset{q(\mathbf{x}_t|\mathbf{x}_0)}{\mathbb{E}}[\mathbb{I}\left(|\hat{\mathbf{x}}_0 - \mathbf{x}_0| > \delta\right)]_{\substack{adaptation}} + \underset{q(\mathbf{y}_t|\mathbf{y}_0)}{\mathbb{E}}[\mathbb{I}\left(|\hat{\mathbf{y}}_0 - \mathbf{y}_0| < \delta\right)]_{\substack{reconstruction}}\right) \tag{4}$$

*where $\hat{\mathbf{x}}_0 = p_\theta(\mathbf{x}_{t:T})$, $\hat{\mathbf{y}}_0 = p_\theta(\mathbf{y}_{t:T})$, indicator function $\mathbb{I}(\cdot)$ and a given threshold $\delta$.*

Specifically, $\mathcal{M}(x_0, y_0, t)$ measures the target domain adaptation degree by assessing how well a noised sample $\mathbf{y}_t$ obtained from $\mathbf{y}_0$ is reconstructed and how likely a noised sample $\mathbf{x}_t$ obtained from $\mathbf{x}_0$ is falsely reconstructed. In the context of the FSIG task, the source domain and target domain differ in attribute, we can assume that, initially, the target domain adaptation degree is close to 0, as the model is trained solely on the source domain. To increase the adaptation degree and enable effective generation in the target domain, we apply the conditioning mechanism for diffusion models.

Few-Shot Image Generation can be considered as a fine-grained conditional generating. Specifically, a conditional generative model can be formulated as $p_t(\mathbf{x}_t \mid \mathbf{y})$, where $\mathbf{y}$ is the condition (given samples in FSIG task). Per Bayes' theorem, $p_t(\mathbf{x}_t \mid \mathbf{y}) \propto p_t(\mathbf{x}_t) p_t(\mathbf{y} \mid \mathbf{x}_t)$. Expressing this relationship as a score function, a score-based conditional diffusion model is described as:

$$\nabla_{\mathbf{x}_t} \log p_t(\mathbf{x}_t \mid \mathbf{y}) = \nabla_{\mathbf{x}_t} \log p_t(\mathbf{x}_t) + \nabla_{\mathbf{x}_t} \log p(\mathbf{y} \mid \mathbf{x}_t) \tag{5}$$

where $\nabla_{\mathbf{x}_t} \log p_t(\mathbf{x}_t)$ and $\nabla_{\mathbf{x}_t} \log p(\mathbf{y} \mid \mathbf{x}_t)$ are respectively the scores of an unconditional DM and a time-dependent intermediate state $(x_t)$ classifier. However, the distribution of $x_t$ at different timestep of diffusion model is different, therefore raising the difficulty of training the classifier. To mitigate overfitting under few-shot, instead of choosing classifier with a simple structure, we propose replacing the time-dependent intermediate state classifier with a non-adaptive Invariant Gradient Matrix $\mathbf{G}(t)$. This matrix captures the essential characteristics of the target domain at each timestep $t$, without relying on the current state $\mathbf{x}_t$. Incorporating $\mathbf{G}(t)$ into the score function (Eq. 5), we obtain:

$$\nabla_{\mathbf{x}_t} \log p_t(\mathbf{x}_t \mid \mathbf{y}) = \nabla_{\mathbf{x}_t} \log p_t(\mathbf{x}_t) + \mathbf{G}(t) \tag{6}$$

The Invariant Gradient Matrix (IGM) $\mathbf{G}(t)$ guides the sampling process towards the target domain, effectively capturing essential domain characteristics under few-shot setting while avoiding overfitting. The training loss associated with our definition of Target Domain Adaptation is defined as:

$$\mathcal{L}_{DA} = \mathop{\mathbb{E}}_{t, \mathbf{x}_0 \in \mathcal{X}, \mathbf{y}_0 \in \mathcal{Y}} \left[ \left| \mathbf{y}_0 - \hat{p}_\theta \left( \sqrt{\bar{\alpha}_t} \mathbf{y}_0 + \sqrt{1 - \bar{\alpha}_t} \epsilon, t \right) \right| - \left| \mathbf{x}_0 - \hat{p}_\theta \left( \sqrt{\bar{\alpha}_t} \mathbf{x}_0 + \sqrt{1 - \bar{\alpha}_t} \epsilon, t \right) \right| \right] \tag{7}$$

where $\hat{p}_\theta$ is the pretrained diffusion model with our IGM, $|\cdot|$ denotes a distance metric.

### 3.3 THEORETICAL ANALYSIS

We shall now provide the theoretical justification of our proposed method.

**IGM Fundamentals** Without loss of generality, let us consider the case at time $t$. To simplify the notation, we denote $\mathbf{c} = \mathbf{G}(t)$. According to Eq. 5 and 6, we have $\mathbf{c} = \nabla_{\mathbf{x}} \log p(\mathbf{y} \mid \mathbf{x}_t)$, solving this differential equation yields $p(\mathbf{y} \mid \mathbf{x}_t) \propto \exp(\mathbf{c} \cdot \mathbf{x}_t)$. This equation defines a pixel-wise linear regression model followed by a softmax activation function, where each pixel of the intermediate state $\mathbf{x}_t$ is weighted by the corresponding element of the IGM. Intuitively, the IGM functions as an attention mechanism that determines how much "attention" or "importance" should be assigned to different regions of $\mathbf{x}_t$, conditioned on a specific target domain $\mathcal{Y}$. See Fig. 1 for an IGM visualization and Section C.1 for further explanation and more visual examples of IGM.

**Overfitting Mitigation Strategy** From a pixel-wise perspective, if each image of the target domain is assigned a unique IGM, it may lead to overfitting as the model can memorize the specific pixel. However, when an IGM is shared across multiple images, it effectively becomes a linear regression model fitting multiple data points, promoting better generalization. To balance model expressiveness and generalization, we introduce the IGM Sharing Degree, $\gamma$, representing the number of images that share an IGM. As $\gamma$ increases from 1, the model shifts from potential overfitting toward better generalization, allowing for fine-tuned performance across diverse datasets. However, excessively high $\gamma$ values can lead to underfitting. We provide an in-depth analysis of this trade-off in Sec. 4.1.

**Theoretical Foundation of Domain Adaptation with IGM** We develop a theoretical framework for domain adaptation in diffusion models, showing how our Invariant Gradient Matrix (IGM) guides the generative process from source domain to target domains towards the desired distribution.

**Theorem 1.** *Let $\mathbf{x}$ be a random variable following a normal distribution with mean $\boldsymbol{\mu}$ and standard deviation $\boldsymbol{\sigma}$. If the conditional probability $p(\mathbf{y} \mid \mathbf{x})$ has the form $p(\mathbf{y} \mid \mathbf{x}) \propto \exp(c \cdot \mathbf{x})$, where $c$ is a constant, then the conditional probability $p(\mathbf{x} \mid \mathbf{y})$ is also a normal distribution, and its posterior density is given by (See Section C.2 for the proof):*

$$p(x|y) = \frac{1}{p(y)\sqrt{2\pi\sigma^2}} \exp\left( -\frac{(x - (\mu + c\sigma^2))^2}{2\sigma^2} + \frac{c^2\sigma^2}{2} + c\mu \right) \tag{8}$$

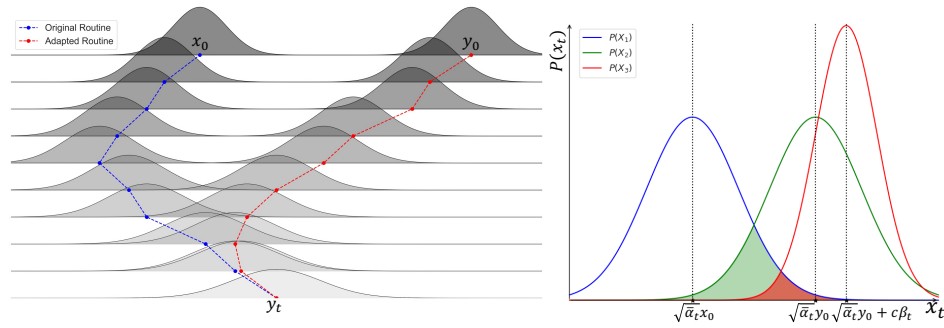

Figure 2: **Left**: Left: A density ridgeline plot showing an 1-D example of our method, transforming a standard normal distribution to a target distribution through an adapted diffusion process. **Right**: Zooming in a specific step from the left plot, the PDFs of $x_t$ (blue), $y_t$ (green) and adapted target domain sample $y_t'$ (red) are shown. The adapted version reduces the overlapping area (green $\to$ red).

**Remark 1.** *According to Theorem 1, the conditional probability $p(\mathbf{x} \mid \mathbf{y})$ differs from the original distribution $p(\mathbf{x})$ in the following aspects:*

1. *Mean shift: The mean of the conditional probability shifts from the original mean $\boldsymbol{\mu}$ to $\mu + c\sigma^2$. This implies that the center of the distribution moves in the direction of $c$, and the distance of the shift is determined by the magnitude of $\boldsymbol{\sigma}$.*

2. *Scaled distribution height: The distribution is vertically scaled at each point by a factor of $\frac{1}{p(y)} \exp\left(\frac{c^2\sigma^2}{2} + c\mu\right)$, based on the observed data and the original hyper-parameters.*

For any samples $x_0 \in \mathcal{X}$ and $y_0 \in \mathcal{Y}$, Eq. 1 defines a forward process in which $x_t$ and $y_t$ progressively approach $\mathcal{N}(\mathbf{0}, \mathbf{I})$. This process ensures that samples from different domains converge to a common Gaussian distribution. The shared endpoint guarantees an overlap between the distributions of $x_t$ and $y_t$ at certain timesteps, despite the model not being trained on the target domain. Conversely, the reverse process starts from $\mathcal{N}(\mathbf{0}, \mathbf{I})$ and aims to recover the training samples. The Fokker-Planck equation (Risken, 1996) describes the evolution of probability density during this process:

$$\frac{\partial p(x,t)}{\partial t} = -\nabla_x \cdot (p(x,t)\nabla_x \log p(x,t)) + \frac{1}{2}\nabla_x^2 p(x,t) \tag{9}$$

The score function $\nabla_x \log p(x,t)$ learned by the model primarily captures the distribution of the source domain $\mathcal{X}$. Consequently, during the reverse process, this source domain-biased score function influences both $x_t$ and $y_t$, causing the generated samples to gravitate towards the source domain distribution, even if $y_t$ has already deviated from its intended trajectory. Intuitively, the learned probability flow acts as a "force" pulling samples towards the center of the source domain $\mathcal{X}$. Our proposed Invariant Gradient Matrix acts as a "counterforce", steering the reverse process towards the target domain while mitigating influence from the source domain. A visual illustration is shown in Fig. 2. For more theoretical analysis from the probability flow point of view refer to Section C.3.

### 3.4 OPTIMIZATION

In Section 3.3, we theoretically analyzed the feasibility of our proposed method. While leveraging a model trained on a source domain that closely resembles the target domain somewhat reduces the complexity of the task, employing a non-adaptive gradient matrix to generate out-of-distribution images still poses significant challenges. Therefore, in this section, we introduce two optimization strategies to further enhance the performance and generalization capability in the target domain.

**Percentile Gradient Clipping** The gradient matrix $\mathbf{G}(t)$ may contain gradient values $g_{i,j}(t)$ at certain pixels that represent noise or weakly correlated information between the source domain $\mathcal{X}$ and the target domain $\mathcal{Y}$. Accordingly, we introduce Percentile Gradient Clipping (PGC) as:

$$\hat{g}_{i,j}(t) = g_{i,j}(t) \cdot (|g_{i,j}(t)| \geq Q(\mathbf{G}(t), \rho)) \tag{10}$$

where $Q(\mathbf{G}(t), \rho)$ represents the $\rho$-th percentile of the gradient matrix $\mathbf{G}(t)$. PGC removes smaller gradients that are more likely to represent noise or weak correlations, while retaining stronger gradi-

ents potentially more informative for target domain adaptation. From an information-theoretic perspective, this process increases the ratio of effective information $\frac{I(\mathbf{G}(t);\mathcal{Y})}{H(\mathbf{G}(t))}$ in $\mathbf{G}(t)$. Here, $I(\mathbf{G}(t);\mathcal{Y})$ represents the mutual information between $\mathbf{G}(t)$ and $\mathcal{Y}$ and $H(\mathbf{G}(t))$ denotes the entropy of $\mathbf{G}(t)$, quantifying its informational uncertainty. Enhancing this ratio enables $\mathbf{G}(t)$ to more effectively capture common features across different domains (Ganin et al., 2016), potentially leading to better generalization in the target domain. For more detail and theoretical analysis see Section C.4.

**Simplified Loss Function** In Section 3.2, according to the definition of Target Domain Adaptation, we can express the domain adaptation loss function as Eq. 7, which aims to encourage the model to reverse intermediate states to the target domain $\mathcal{Y}$ instead of the source domain $\mathcal{X}$. However, experimental results suggest that this approach may suppress useful knowledge learned by the model in the source domain. Considering that the goal of FSIG is to select a source domain $\mathcal{X}$ that is close to the target domain $\mathcal{Y}$, we can simplify the loss function by emphasizing reconstruction ability:

$$\mathcal{L}_{DA} = \mathop{\mathbb{E}}_{\mathbf{y}_0 \in \mathcal{Y}} \left[ \left| \mathbf{y}_0 - \hat{p}_\theta \left( \sqrt{\bar{\alpha}_t} \mathbf{y}_0 + \sqrt{1 - \bar{\alpha}_t} \epsilon, t \right) \right| \right] \tag{11}$$

This simplified loss function allows the model to retain useful knowledge learned from the source domain while adapting to the target domain. Intuitively, by minimizing the reconstruction error of target domain samples, the model naturally gravitates towards the target domain while preserving relevant information from the source domain to the greatest extent possible.

## 4 EXPERIMENTS

**Datasets and Baseline** Following previous work (Wang et al., 2018; Li et al., 2020; Ojha et al., 2021), we used Flickr Faces HQ (FFHQ) (Karras et al., 2019) as the source domain datasets for all quantitative analysis, LSUN (Yu et al., 2015) and FFHQ for qualitative analysis. We applied our method to adapt to the following common target domains for comparisons to existing FSIG methods: FFHQ-Babies (Ojha et al., 2021), FFHQ-Sunglasses (Ojha et al., 2021), MetFaces (Karras et al., 2020), portrait paintings from the artistic faces dataset (Yaniv et al., 2019). We select three FSIG methods as baseline, including RICK (SOTA method) (Zhao et al., 2023), GenDA (SOTA representation learning method for GAN) (Mondal et al., 2022), CRDI (SOTA representation learning method for DM) (Cao & Gong, 2024). More methods comparison results are given in Section F.

**Metrics** We compute two commonly used metrics in FSIG, FID (Fréchet inception distance) (Heusel et al., 2017) and Intra-LPIPS (Intra-cluster pairwise Learned Perceptual Image Patch Similarity) Ojha et al. (2021), to quantitatively assess the quality and diversity of generated samples with respect to the target domain, respectively. We also calculate MC-SSIM (Mode Coverage Structural Similarity Index Measure) (Cao & Gong, 2024) which quantify the mode coverage for complex domain.

**Implementation Details** We used Guided Diffusion (Dhariwal & Nichol, 2021) framework from OpenAI and pretrained weight from Segmentation DDPM (Baranchuk et al., 2021). We utilized DDIM (Song et al., 2020a) with 25 inference steps to improve the efficiency while training. Model training is performed with 256 x 256 resolution and batch size 10 on a single A100/H100 GPU.

### 4.1 SHARING DEGREE: BALANCING GENERALIZATION AND SPECIFICITY

To validate the theoretical analysis presented in Sec. 3.3 regarding the impact of the IGM sharing degree on overfitting, we conducted experiments across three commonly used target domains in FSIG, Babies, Sunglasses and MetFaces. We applied our method for each domain at three different timestep periods $[t_s, t_e]$ during the diffusion process, varying the degree of IGM sharing. We evaluated the generated images using FID scores; the results are shown in Fig. 3. The IGM sharing degree, $\gamma$, ranges from 1 (one IGM per one image) to 10 (one IGM per ten images). We additionally fitted an Exponential Moving Average (EMA) curve (green line) to each graph to highlight the overall trend.

It can be observed that, for the target domain Babies and Sunglasses, the EMA of FID shows varying degrees of the U-shaped curve as the IGM sharing degree increases from 1 to 10. When $\gamma = 1$, the model exhibits the highest degree of overfitting, resulting in images generated with low diversity. As shown in Fig. 4a (middle), some modifications are concentrated on facial expressions without altering personal identity. As $\gamma$ increases, the FID ($\downarrow$) decreases, reaching a minimum at an optimal sharing degree. This optimum balances specific image feature capture and generalizable pattern learning of a

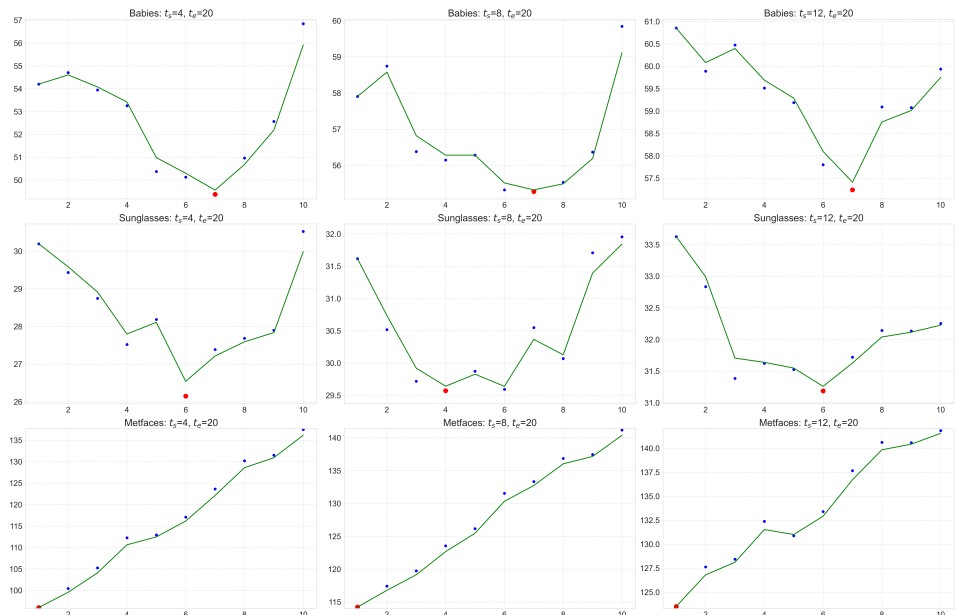

Figure 3: FID (↓) values across different IGM sharing degrees (γ) for three target domains in FSIG: Babies, Sunglasses, and MetFaces. Each subplot represents a different domain, where the x-axis denotes the IGM sharing degree γ, ranging from 1 (one IGM per one image) to 10 (one IGM per ten images), and the y-axis shows the corresponding FID score. An Exponential Moving Average curve (green line) illustrates the trend, and the lowest FID(↓) score is marked with a red dot.

target domain. Further increasing $\gamma$ to 10 leads to FID (↓) increase, indicating underfitting. Generated images include source domain samples due to insufficient fitting capacity. IGMs fail to fully adapt the source model to the target domain, as the orange boxed samples in Fig. 4a (right). This phenomenon illustrates the trade-off between model capacity and generalization in IGM-guided DMs.

In contrast, the MetFaces target domain exhibits a distinct pattern. When $\gamma = 1$, generated samples closely match the target domain style but lack diversity. As the sharing degree increases to 10, the generated samples predominantly resemble the source domain, with only slight characteristics of the target domain (Fig. 4a second row, right). This behavior differs from Babies and Sunglasses, where intermediate sharing degrees yield optimal results. For MetFaces, the significant disparity from the source domain exposes the limitations of IGMs in bridging large domain gaps, resulting in effective target domain capture only at lower sharing degrees (We provide a detailed analysis in Sec. 4.3 and visualization in Section C.1). This finding highlights the importance of selecting an appropriate source domain that shares sufficient similarities with the target domain in FSIG tasks.

## 4.2 MAIN RESULTS ON FSIG

Building upon the insights from our analysis of IGM sharing degree, we now apply our method to real-world Few-Shot Image Generation (FSIG) experiments. In this section, we present a comparative evaluation of our approach against current state-of-the-art (SOTA) methods; the quantitative results are shown in Tab. 1. To demonstrate the robustness of our method, we further present the experimental results with sharing degrees of 10 (FS-DRL-10) and 5 (FS-DRL-5). These configurations utilize one-tenth and one-half of the parameters employed in the CRDI (Cao & Gong, 2024), respectively.

As seen in Tab. 1, FS-DRL significantly improves the performance of representation learning method in FSIG. However, in Babies and MetFaces, a gap remains compared to fine-tuning methods in terms of FID. Consistent with the findings of Cao & Gong (2024), we observe that while fine-tuning approaches achieve better performance on evaluation metrics, they tend to produce samples with certain visual artifacts. In contrast, representation learning methods generate "cleaner" samples, but with reduced diversity. See Fig. 4b for visual examples. However, FID score failed to capture these differences, as in Fig. 3 (first and second rows), FID scores at $\gamma = 1$ and 10 are comparable.

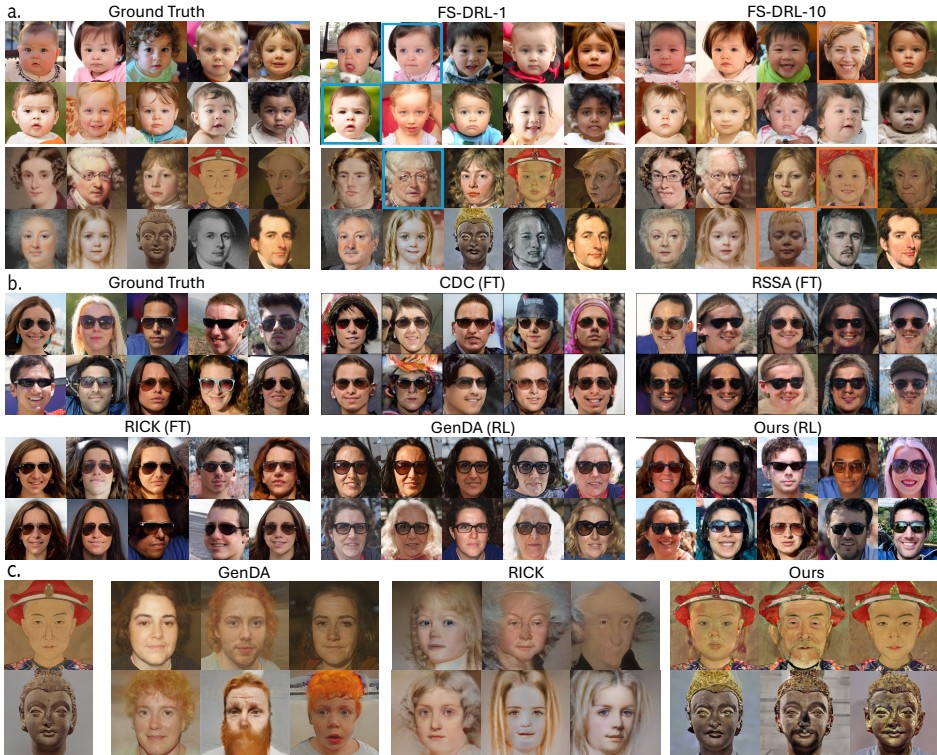

Figure 4: **a**: Impact of IGM sharing degree (FS-DRL-$\gamma$) on generated image quality and diversity, highlighting source domain leakage (orange) and low diversity (blue) (First row: Babies, Second row: MetFaces). **b**: Visual examples of our method alongside four other high-performance methods on Sunglasses (RL: Representation Learning and FT: Fine-Tuning). **c**: Mode coverage comparison across GenDA, RICK and our method. For each row, the leftmost image is from the MetFaces target domain, followed by the most similar (SSIM) generated images. Please zoom in for more details.

Table 1: Comparing FID ($\downarrow$) Scores and MC-SSIM ($\uparrow$) (for MetFaces only) between our methods and the baselines (Mean $\pm$ Std.). FS-DRL-$\gamma$ represents our method with a sharing degree $\gamma$, and FS-DRL-opt denotes the optimized result. RL and FT represent Representation Learning and Fine-Tuning, respectively. Best in **bold** and the second best in **underline with bold**.

| | | **Babies** | **Sunglasses** | **MetFaces** | |
|---|---|---|---|---|---|
| **Method** | **Type** | FID $\downarrow$ | FID $\downarrow$ | FID $\downarrow$ | MC-SSIM $\uparrow$* |
| GenDA | RL | $63.31 \pm 0.05$ | $35.64 \pm 0.15$ | $104.48 \pm 0.58$ | $0.33 \pm 0.03$ |
| RICK | FT | $\mathbf{39.39} \pm 0.09$ | $25.22 \pm 0.11$ | $\mathbf{48.53} \pm 0.34$ | $0.41 \pm 2e\text{-}3$ |
| CRDI | RL | $48.52 \pm 0.28$ | $24.62 \pm 0.18$ | $94.86 \pm 0.72$ | $0.62 \pm 5e\text{-}3$ |
| FS-DRL-10 | RL | $56.96 \pm 0.31$ | $31.69 \pm 0.25$ | $110.54 \pm 0.50$ | $0.57 \pm 0.01$ |
| FS-DRL-5 | RL | $43.73 \pm 0.29$ | $\underline{\mathbf{22.69}} \pm 0.16$ | $88.36 \pm 0.52$ | $\underline{\mathbf{0.64}} \pm 7e\text{-}3$ |
| FS-DRL-opt | RL | $\underline{\mathbf{41.95}} \pm 0.22$ | $\mathbf{21.93} \pm 0.16$ | $\underline{\mathbf{77.17}} \pm 0.43$ | $\mathbf{0.70} \pm 2e\text{-}3$ |

*Calculated using 5000 samples for improved stability compared to prior work.

This indicates the limitation of FID score in distinguishing between source domain leakage and low diversity issues, as it measures both quality and diversity using feature space distances.

This limitation is particularly evident in complex domains like MetFaces (given samples in Fig.4a second row, left). While fine-tuning methods achieve lower FID ($\downarrow$) scores, they capture only a limited subset of styles with prominent artifacts. Our approach, despite higher FID ($\downarrow$) scores, achieve superior mode coverage and sample quality. To better quantify this aspect, we employ the MC-SSIM metric (Tab.1 last column), which shows that our method outperforms others in preserving target domain styles. Fig.4c provides qualitative results of this advantage. These findings underscore the importance of using complementary metrics for comprehensive model evaluation in FSIG tasks and highlight the strength of our approach in maintaining target domain styles.

Table 4: Comparison of model performance under 10-shot, 5-shot, and 1-shot with GenDA and CRDI, evaluated based on generation quality using the FID score ($\downarrow$). Best in **Bold**.

| | 1-shot | | 5-shot | | 10-shot | |
|---|---|---|---|---|---|---|
| **Methods** | **Babies** | **Sunglasses** | **Babies** | **Sunglasses** | **Babies** | **Sunglasses** |
| GenDA | 105.13 | 83.70 | 65.47 | 45.44 | 62.14 | 35.64 |
| CRDI | 100.85 | 74.60 | 55.87 | 31.35 | 48.52 | 24.62 |
| Ours | **95.90** | **60.99** | **48.27** | **28.45** | **41.95** | **21.93** |

## 4.3 FURTHER ANALYSIS AND DISCUSSION

**Effective of Percentile Gradient Clipping** Tab. 2 demonstrates the impact of Percentile Gradient Clipping (PGC) across three target domains. The results show a U-shaped trend in FID scores, indicating the presence of noise in the IGM that can be effectively removed using PGC. However, excessive

Table 2: Comparisons of model performance with different $\rho$, evaluated by the FID ($\downarrow$). Best in **Bold**.

| $\rho$-th | 0 | 20 | 40 | 60 | 80 |
|---|---|---|---|---|---|
| Babies | 45.70 | 44.56 | 42.40 | **41.95** | 43.53 |
| Sunglasses | 22.46 | 22.08 | **21.93** | 22.55 | 25.90 |
| MetFaces | 78.31 | **77.17** | 79.38 | 81.66 | 88.19 |

clipping eliminates informative gradients, degrading results. For Babies and Sunglasses, performance improves significantly with high percentile clipping (40th-60th), which suggests that IGM for these domains is inherently sparse. Conversely, MetFaces performs optimally at a lower percentile (20th), implying a denser IGM that requires more gradient information preservation; see the visualization and in-depth analysis in Section C.1. These divergent behaviors highlight IGM adaptability to domain complexity, motivating further exploration of domain-specific parameter optimization techniques.

**Further Decrease Number of Parameter** To explore the possibility of further reducing the number of parameters in our Invariant Gradient Matrix (IGM), we investigated two additional approaches: Upsampling and Low-Rank Matrix Approximation (LRMA). For Upsampling, we initialize a low-resolution gradient matrix $\mathbf{G}_{low}(t) \in \mathbb{R}^{m \times m}$, where $m < n$, with $n$ being the dimen-

Table 3: Comparisons of model performance and parameter count when further decrease number of parameter using Upsampling and LRMA, evaluated by the FID ($\downarrow$). Best in **Bold**.

| | Upsampling | | LRMA | | Original |
|---|---|---|---|---|---|
| # **Params** | $m$=64 12$K$ | $m$=128 49$K$ | $r$=64 37$K$ | $r$=128 82$K$ | $n$=256 196$K$ |
| Babies | 58.45 | 54.53 | 45.96 | 43.21 | **41.95** |
| Sunglasses | 33.16 | 30.62 | 40.21 | 39.89 | **21.93** |
| MetFaces | 100.46 | 88.21 | 133.42 | 131.49 | **77.17** |

sionality of the input samples. During the training and sampling process, we upsample $\mathbf{G}_{low}(t)$ to the original resolution using bilinear interpolation. For LRMA, we assume that $\mathbf{G}(t) = \mathbf{U}(t)\mathbf{\Sigma}(t)\mathbf{V}(t)^T$ is an anti-symmetric matrix, where $\mathbf{U}(t) \in \mathbb{R}^{n \times r}, \mathbf{\Sigma}(t) \in \mathbb{R}^{r \times r}, \mathbf{V}(t) \in \mathbb{R}^{n \times r}$, with $r < n$. The results are shown in Tab. 3. These results indicate that, while IGM exhibits some sparsity, simple parameter reduction methods may not effectively capture its full information content. LRMA shows more promise, particularly on certain datasets, but requires further refinement to achieve performance comparable to that of the original method across diverse datasets.

**From Few-Shot to One-Shot** To evaluate the performance of our method in more extreme scenarios, we designed experiments under 5-shot and 1-shot settings. In these cases, conventional models face an increased risk of overfitting. However, our approach, leveraging the adjustable sharing degree $\gamma$, demonstrates significant advantages. As shown in Tab. 4, our method significantly outperforms GenDA (Mondal et al., 2022) and CRDI Cao & Gong (2024) under both 5-shot and 1-shot scenarios, highlighting its effectiveness in extreme few-shot conditions.

## 5 CONCLUSION

We present a novel representation learning framework for Few-Shot Image Generation, featuring a tunable parameter to explicitly mitigate overfitting while adapting a specific domain. Our method achieves competitive SOTA performance while surpassing representation learning-based approaches using only half of the parameters. By focusing on the diffusion process, our approach is compatible with all diffusion models, offering a versatile and efficient solution for Few-Shot Image Generation.

## REPRODUCIBILITY STATEMENT

To ensure that the proposed work is reproducible, we have included a pseudocode for training (Algo. 1) and sampling (Algo. 2). We have an explicit section (Sec. 4) with implementation details. We have also clearly mentioned evaluation details in Section .E. Complete code will be released upon acceptance.

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

## A    APPENDIX

This is the appendix for "Exploring Few-Shot Image Generation With Minimized Risk of Overfitting". Tab. 5 summarizes the abbreviations and symbols used in the paper.

This appendix is organized as follows:

- Section B discusses the limitation and broader impact of our work.
- Section C gives the full proof of our Theorem with additional explanation.
- Section D presents additional details of our approach.
- Section E presents additional details of the FSIG evaluation metric.
- Section F presents additional quantitative and qualitative results.

Table 5: List of abbreviations and symbols used in the paper

| Abbreviation/Symbol | Meaning |
|---|---|
| **Abbreviation** | |
| Sec. A.B | Section in the main paper |
| Section. A.B | Section in the Appendix |
| FSIG | Few-Shot Image Generation |
| DM | Diffusion Model |
| DDPM | Denoising Diffusion Probabilistic Model |
| IGM | Invariant Gradient Matrix |
| LRMA | Low-Rank Matrix Approximation |
| **Symbol in Theory** | |
| $\mathcal{X}$ | Source Domain |
| $\mathcal{Y}$ | Target Domain |
| $G(t)$ | Invariant Gradient Matrix |
| $g_{i,j}(t)$ | $(i,j)$-th element of Invariant Gradient Matrix $G(t)$ |
| $\mathcal{M}(\cdot)$ | Domain adaptation measure |
| **Symbol in Algorithm** | |
| $x_0$ | Original source domain sample |
| $x_t$ | Noisy original source sample after $t$ forward step |
| $y_0$ | Target domain sample |
| $y_t$ | Noisy target sample after $t$ forward step |
| $q(\cdot)$ | Distribution in the encoding process |
| $p_\theta(\cdot)$ | Distribution in the $\theta$-parameterized decoding process |
| $\rho$ | Percentile of percentile gradient clipping |
| $\hat{p}_\theta$ | Pretrained diffusion model with our IGM |
| $\theta$ | Parameter of U-Net |
| $\hat{x}_0$ | Reconstructed source domain sample $x_0$ |
| $\hat{y}_0$ | Reconstructed target domain sample $y_0$ |
| $T$ | Total time-steps |
| $\beta_1, \ldots, \beta_T$ | Variance schedule |
| $\alpha_t$ | $1 - \beta_t$ |
| $\bar{\alpha}_t$ | $\prod_{s=1}^{t} \alpha_s$ |

## B    LIMITATION AND BROADER IMPACT

**Limitation**    Although our method effectively balances specificity and generalization, its performance degrades when the disparity between the source and target domains is substantial, such as MetFaces (Karras et al., 2020). In such cases, overfitting tends to outperform underfitting (Fig. 3). A potential solution involves incorporating Large Multi-modality Models (LMMs) like CLIP (Radford et al., 2021) to constrain style more effectively, allowing the Invariant Gradient Matrix to preserve more non-style information. We avoided using CLIP to minimize target domain exposure, as LMMs may have been trained on these samples. However, if this constraint can be relaxed, integrating

LMMs could enhance our method's robustness across diverse domains. Future work will explore this integration while maintaining data privacy.

**Broader Impact**  Although our method outperforms state-of-the-art (SOTA) approaches in various comparisons, our research is not centered on topping leaderboards but rather on exploring the limits of FSIG while "fundamentally" avoiding overfitting. It is worth noting that while diffusion models have made impressive progress in recent years, surpassing GANs in most fields, they are rarely used in Few-Shot Image Generation (FSIG) tasks. This is primarily because most FSIG methods rely on fine-tuning, and diffusion models, despite being trained on the same datasets, have more parameters, making them seemingly "unsuitable" for FSIG tasks.

However, on the one hand, the training data for large models continues to expand rapidly and is becoming crucial in many real-world applications. On the other hand, although large models can generate highly realistic images, they still underperform on most user-defined real-world subjects. This gap requires FSIG methods that can align with the capabilities of these large models. Our method presents a novel attempt toward this goal, showing promising initial progress.

## C  PROOF AND ADDITIONAL THEORETICAL ANALYSIS

### C.1  ADDITIONAL ANALYSIS OF EQUIVALENT CLASSIFIER

Consider the gradient of the log-conditional probability:

$$\nabla_{\mathbf{x}} \log p(\mathbf{y} \mid \mathbf{x}_t) = \mathbf{c} \tag{12}$$

This differential equation can be solved to obtain the form of $p(\mathbf{y} \mid \mathbf{x}_t)$. Integrating both sides with respect to $\mathbf{x}$:

$$\int \nabla_{\mathbf{x}} \log p(\mathbf{y} \mid \mathbf{x}_t) \cdot d\mathbf{x} = \int \mathbf{c} \cdot d\mathbf{x} \tag{13}$$

yield

$$\log p(\mathbf{y} \mid \mathbf{x}_t) = \mathbf{c} \cdot \mathbf{x}_t + K \tag{14}$$

where $K$ is an integration constant. Exponentiating both sides:

$$p(\mathbf{y} \mid \mathbf{x}_t) = \exp(\mathbf{c} \cdot \mathbf{x}_t + K) = \exp(K) \cdot \exp(\mathbf{c} \cdot \mathbf{x}_t) \tag{15}$$

Let $Z = \exp(K)$, which serves as a normalization constant. Thus:

$$p(\mathbf{y} \mid \mathbf{x}_t) = Z \cdot \exp(\mathbf{c} \cdot \mathbf{x}_t) \tag{16}$$

This exponential form aligns with the softmax mechanism, where $\mathbf{c}$ acts as an attention matrix, determining the "attention" or "importance" of different regions in the state space given $\mathbf{y}$.

**Invariant Gradient Matrix Visualization**  To validate our theoretical analysis, we visualized the Invariant Gradient Matrices (IGMs) at different diffusion timesteps for three target domains: Babies, Sunglasses, and MetFaces (Fig. C.1). Notably, for Babies and Sunglasses domains, the IGMs exhibit significant sparsity, aligning with our analysis in Sec.4.3. In contrast, the IGM for MetFaces contains more intricate details, likely capturing additional information such as style variations. This increased complexity in the MetFaces IGM correlates with the observed reduction in diversity, as the model focuses on preserving more domain-specific features.

### C.2  PROOF OF THEOREM 1 AND REMARK 1

Let $x$ be a random variable following a normal distribution, $\mathcal{N}(\mu, \sigma)$, i.e.,

$$p(x) = \frac{1}{\sqrt{2\pi\sigma^2}} \exp\left(-\frac{(x-\mu)^2}{2\sigma^2}\right) \tag{17}$$

Assume that the conditional probability $p(y|x)$ has the form:

$$p(y|x) = \exp(cx) \cdot \text{const} \tag{18}$$

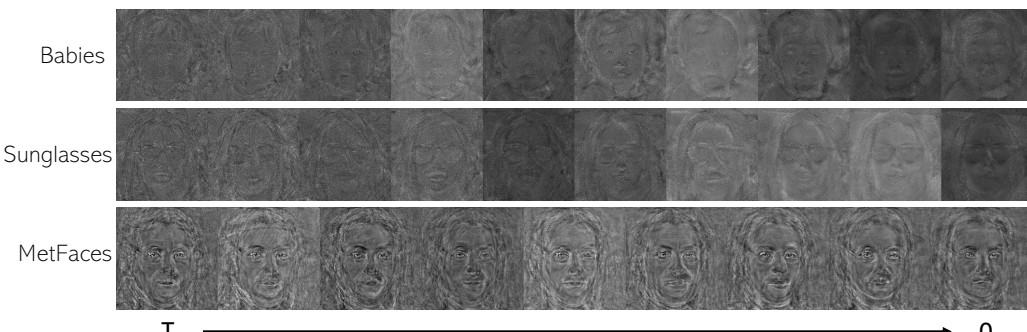

Figure 5: Visualization of Invariant Gradient Matrices (IGMs) across three target domains: Babies, Sunglasses, and MetFaces. Each row represents the IGM at different diffusion timesteps for the corresponding domain.

where $c$ is an invariant variable (IGM in our case). Applying Bayes' theorem, we obtain (the constant term from $p(y|x)$ is absorbed into $p(y)$):

$$p(x|y) = \frac{p(y|x)p(x)}{p(y)} = \frac{\exp(cx)}{p(y)\sqrt{2\pi\sigma^2}} \exp\left(-\frac{(x-\mu)^2}{2\sigma^2}\right) \tag{19}$$

Combining the exponential terms, we have:

$$p(x|y) = \frac{1}{p(y)\sqrt{2\pi\sigma^2}} \exp\left(cx - \frac{(x-\mu)^2}{2\sigma^2}\right) \tag{20}$$

By completing the square, we can rewrite the expression as:

$$p(x|y) = \frac{1}{p(y)\sqrt{2\pi\sigma^2}} \exp\left(-\frac{(x-(\mu+c\sigma^2))^2}{2\sigma^2} + \frac{c^2\sigma^2}{2} + c\mu\right) \tag{21}$$

This expression shows that $p(x|y)$ is also a normal distribution with mean $\mu + c\sigma^2$ and variance $\sigma^2$, where the normalization constant is given by:

$$\frac{1}{p(y)} \exp\left(\frac{c^2\sigma^2}{2} + c\mu\right) \tag{22}$$

For a $d$-dimensional case where all dimensions are independent, we can treat each dimension separately and combine the results. The mean of each dimension will be updated as $\mu_i + c_i\sigma_i^2$, where $i$ is the dimension index. The variances remain unchanged. The overall normalization constant will be the product of the normalization constants for each dimension. $\square$

### C.3 THEORETICAL ANALYSIS OF PROBABILITY FLOW CORRECTION

Consider a diffusion model with probability density function (PDF) $p(x, t)$ for its data distribution, where $x$ represents the data and $t$ represents the time step of the diffusion process. The probability flow vector field $v(x, t)$ satisfies the modified Fokker-Planck equation with a diffusion coefficient $g(t)$:

$$\frac{\partial p(x,t)}{\partial t} = -\nabla_x \cdot (p(x,t)v(x,t)) + \frac{1}{2}\nabla_x \cdot \left(g(t)^2 \nabla_x p(x,t)\right). \tag{23}$$

The first term $-\nabla_x \cdot (p(x,t)v(x,t))$ represents the drift induced by the vector field $v(x,t)$, while the second term $\frac{1}{2}\nabla_x \cdot \left(g(t)^2 \nabla_x p(x,t)\right)$ accounts for diffusion, with $g(t)$ ($\sqrt{\beta_t}$ in DDPM) as the time-dependent diffusion coefficient.

To improve the alignment of the model's probability flow with the target domain, we introduce a correction term $\delta v(x, t)$:

$$\hat{v}(x,t) = v(x,t) + \delta v(x,t), \tag{24}$$

where $\delta v(x, t)$ is learned from an underfitted classifier at the intermediate state $t$. This correction term can be represented as:

$$\delta v(x,t) = \mathbb{E}_{\theta_c \sim p(\theta_c|x)}[f(x,\theta_c)], \tag{25}$$

where $\theta_c$ represents the classifier parameters, $p(\theta_c|x)$ is the posterior distribution given the data $x$, and $f(x, \theta_c)$ maps these parameters to a correction in the probability flow. This correction aims to capture the discrepancy between the current model state and the target domain.

By introducing the correction, the modified vector field $\hat{v}(x, t)$ adjusts the dynamics of the diffusion process, resulting in the corrected Fokker-Planck equation:

$$\frac{\partial p(x,t)}{\partial t} = -\nabla_x \cdot (p(x,t)\hat{v}(x,t)) + \frac{1}{2}\nabla_x \cdot \left(g(t)^2\nabla_x p(x,t)\right). \tag{26}$$

**Proof (Informal)** To analyze the effect of the correction $\delta v(x, t)$, we expand the divergence term in the corrected Fokker-Planck equation:

$$\begin{aligned}
\frac{\partial p(x,t)}{\partial t} &= -\nabla_x \cdot (p(x,t)\hat{v}(x,t)) + \frac{1}{2}\nabla_x \cdot \left(g(t)^2\nabla_x p(x,t)\right) \\
&= -\nabla_x \cdot (p(x,t)(v(x,t) + \delta v(x,t))) + \frac{1}{2}\nabla_x \cdot \left(g(t)^2\nabla_x p(x,t)\right) \\
&= -\nabla_x \cdot (p(x,t)v(x,t)) - \nabla_x \cdot (p(x,t)\delta v(x,t)) + \frac{1}{2}\nabla_x \cdot \left(g(t)^2\nabla_x p(x,t)\right).
\end{aligned} \tag{27}$$

The term $-\nabla_x \cdot (p(x,t)v(x,t)) + \frac{1}{2}\nabla_x \cdot \left(g(t)^2\nabla_x p(x,t)\right)$ corresponds to the original diffusion model, while the new term $-\nabla_x \cdot (p(x,t)\delta v(x,t))$ introduces a correction based on the classifier. This correction guides the probability flow to better match the target distribution. $\square$

In summary, by modifying the probability flow vector field to $\hat{v}(x, t)$, we adjust the generative process to produce samples that more closely align with the target data distribution, enhancing both the quality and diversity of the generated samples.

## C.4 THEORETICAL ANALYSIS OF PERCENTILE GRADIENT CLIPPING

Given a gradient matrix $\mathbf{G}(t)$ containing gradient information between the source domain $\mathcal{X}$ and the target domain $\mathcal{Y}$, let $Q(\mathbf{G}(t), \rho)$ denote the $\rho$-th percentile of $\mathbf{G}(t)$. Define the gradient clipping operation $\mathcal{T}$ as follows:

$$\mathcal{T}(\mathbf{G}(t))_{i,j} = \begin{cases} 0, & \text{if } |g_{i,j}(t)| < Q(\mathbf{G}(t), p) \\ g_{i,j}(t), & \text{otherwise} \end{cases} \tag{28}$$

where $g_{i,j}(t)$ denotes the $(i, j)$-th element of $\mathbf{G}(t)$. Then, the gradient clipping operation $\mathcal{T}$ satisfies the following inequality:

$$\frac{I(\mathcal{T}(\mathbf{G}(t)); \mathcal{Y})}{H(\mathcal{T}(\mathbf{G}(t)))} \geq \frac{I(\mathbf{G}(t); \mathcal{Y})}{H(\mathbf{G}(t))} \tag{29}$$

where $I(\cdot; \cdot)$ denotes the mutual information and $H(\cdot)$ denotes the entropy. In other words, the gradient clipping operation $\mathcal{T}$ increases the ratio of effective information, enabling the clipped gradient matrix $\mathcal{T}(\mathbf{G}(t))$ to capture the characteristics of the target domain $\mathcal{Y}$ more effectively.

**Proof (Informal)** The gradient clipping operation $\mathcal{T}$ sets the elements of $\mathbf{G}(t)$ with smaller magnitudes to zero. This is equivalent to removing the gradient information that has a relatively weak influence on the target domain $\mathcal{Y}$. Since elements with smaller magnitudes are assumed to contribute less to mutual information $I(\mathbf{G}(t); \mathcal{Y})$, their removal has a limited impact on the overall mutual information between the gradient matrix and the target domain. At the same time, removing this information reduces the entropy $H(\mathbf{G}(t))$ of $\mathbf{G}(t)$, since it reduces the overall noise and randomness in the gradient matrix.

Specifically, let $g_{i,j}(t)$ denote the $(i, j)$-th element of $\mathbf{G}(t)$. The clipping threshold $Q(\mathbf{G}(t), \rho)$ is selected such that elements below this threshold contribute minimally to the mutual information $I(\mathbf{G}(t); \mathcal{Y})$. Hence, we have:

$$I(\mathcal{T}(\mathbf{G}(t)); \mathcal{Y}) \approx I(\mathbf{G}(t); \mathcal{Y})$$

At the same time, setting these elements to zero reduces the entropy $H(\mathbf{G}(t))$, as the sparsity of $\mathcal{T}(\mathbf{G}(t))$ increases and the overall uncertainty within the gradient matrix is reduced. This reduction in entropy is significant, since the clipped elements are removed entirely, resulting in:

$$H(\mathcal{T}(\mathbf{G}(t))) < H(\mathbf{G}(t))$$

Therefore, the ratio of mutual information to entropy increases after clipping:

$$\frac{I(\mathcal{T}(\mathbf{G}(t)); \mathcal{Y})}{H(\mathcal{T}(\mathbf{G}(t)))} > \frac{I(\mathbf{G}(t); \mathcal{Y})}{H(\mathbf{G}(t))}$$

In essence, the gradient clipping operation $\mathcal{T}$ preserves the information that is relevant to the target domain $\mathcal{Y}$ while reducing the entropy of the gradient matrix. This increases the relative effectiveness of the retained information, allowing $\mathcal{T}(\mathbf{G}(t))$ to more effectively capture the characteristics of the target domain. $\square$

## D  ADDITIONAL DETAIL FOR APPROACH

**Training Algorithm**   Algo. 2 shows the training pseudocode when $\gamma = 10$. When $\gamma < 10$, we randomly create a mapping function to distribute the images such that each IGM may correspond to multiple images. Specifically, as $\gamma$ decreases, we aim to evenly distribute the images among the available IGMs. When $\gamma$ eventually reduces to one, it results in a single IGM corresponding to all images. This mapping approach ensures that the images are distributed fairly and shared as evenly as possible across varying $\gamma$.

---

**Algorithm 1** FS-DRL - Training Pseudo-code

---

1: **Input:** Target Domain $\mathcal{Y} = \{y^0, y^1, ..., y^{n-1}\}$ (n=10), start point $t_s$, end point $t_e$, Randomly Initialized IGM $G_\theta(t)$, a Frozen Noise Network (DM) $\epsilon_\theta$ and Learning Rate $\nu$.
2: **while** not converge **do**
3:     Sample: $t$ uniformly from $[t_s, ..., t_e]$
4:     **for** $i, y^i$ in *enumerate*($\mathcal{Y}$) **do**
5:         Given $y^i_{t-1} \leftarrow$ sample from $\sqrt{\bar{\alpha}_{t-1}} y^i_0 + \sqrt{1 - \bar{\alpha}_{t-1}} \epsilon, \quad \epsilon \sim \mathcal{N}(0, \mathbf{I})$
6:         $\hat{\epsilon} \leftarrow \epsilon_\theta\left(y^i_t\right) - \sqrt{1 - \bar{\alpha}_t} G_\theta(t, i)$
7:         $\hat{y}^i_0 \leftarrow \frac{y^i_t - \sqrt{1 - \bar{\alpha}_t} \hat{\epsilon}}{\sqrt{\bar{\alpha}_t}}$
8:         $G_\theta(t, i) \leftarrow G_\theta(t, i) - \nu \nabla_{G_\theta(t,i)} \mathcal{L} |y^i_0 - \hat{y}^i_0|$
9: **return** $G_\theta$

---

**Sampling Algorithm**  We show the sampling pseudocode in Algo. 2.

---

**Algorithm 2** FS-DRL - Sampling Code

---

1: **Input:** Target Domain $\mathcal{Y} = \{y^0, y^1, ..., y^{n-1}\}$ (n=10), start point $t_s$, end point $t_e$, Proposed IGM $G_\theta$(t), a Frozen Noise Network (DM) $\epsilon_\theta$ and a $mask$ (Percentile Gradient Clipping).
2: Sample $y_0$ randomly from $\mathcal{Y}$, $i$ from [0, ..., n-1], Set $t \leftarrow t_e$
3: $y_t \leftarrow$ sample from $\sqrt{\bar{\alpha}_t} y_0 + \sqrt{1 - \bar{\alpha}_t} \epsilon, \quad \epsilon \sim \mathcal{N}(0, \mathbf{I})$
4: **for** $t$ **in** *reversed(range($t_e + 1$))* **do**
5:     **if** $t < t_s$ **then**
6:         $G_\theta(t, i) \leftarrow 0$
7:     $\hat{\epsilon} \leftarrow \epsilon_\theta\left(y_t\right) - \sqrt{1 - \bar{\alpha}_t} (G_\theta(t, i) \odot mask)$
8:     $\hat{y}_0 \leftarrow \frac{y_t - \sqrt{1 - \bar{\alpha}_t} \hat{\epsilon}}{\sqrt{\bar{\alpha}_t}}$
9:     $y_{t-1} \leftarrow$ sample from $\sqrt{\bar{\alpha}_i} \hat{y}_0 + \sqrt{1 - \bar{\alpha}_i} \epsilon, \quad \epsilon \sim \mathcal{N}(0, \mathbf{I})$
10: **return** $y_0$

---

## E  ADDITIONAL DETAIL FOR EVALUATION

**Implemented Intra-LPIPS Algorithm**  As most implementations of Intra-LPIPS skip empty clusters when calculating the average, reducing the number of comparisons (e.g., from 10 to only 3), misrepresenting true diversity, we modify the implementation as Algo. 3 (modified parts in red).

**Implemented MC-SSIM Algorithm**  For pseudocode of MC-SSIM please refer to Algo. 4. Note that in Tab. 1, MC-SSIM was calculated using 5000 samples for improved stability, which may lead to disparities with prior work.

---

**Algorithm 3** Calculate Intra-LPIPS within clusters

---

1: **Input:**
2: 1. Generated images $X = x_1, \ldots, x_b$
3: 2. Real image dataloader $L$
4: 3. Number of images per cluster $m$
5: **Output:** Average Intra-LPIPS within clusters
6:
7: Step 1. Initialize empty clusters $C_i = \emptyset$ for $i \in 0, \ldots, 9$
8: **for** $i = 1, \ldots, b$ **do**
9:     Initialize distances $D = []$
10:    **for** real image $r$ in $L$ **do**
11:       $d = \texttt{LPIPS}(x_i, r)$             $\triangleright$ Compute LPIPS distance
12:       $D.\text{append}(d)$
13:    $j = \arg\min_j D$               $\triangleright$ Index of closest cluster
14:    $C_j.\text{append}(i)$               $\triangleright$ Assign $x_i$ to cluster $C_j$
15:
16: Step 2. Restrict clusters to size $m$
17: **for** $i = 0, \ldots, 9$ **do**
18:    $C_i = C_i[1 : m]$
19:
20: Step 3. Compute pairwise Intra-LPIPS within each cluster
21: Initialize distances $D = []$
22: **for** $i = 0, \ldots, 9$ **do**
23:    Initialize temp distances $T = [0]$    $\triangleright$ Initialize $T$ with $[0]$ instead of an empty list.
24:    **for** $j = 1, \ldots, |C_i|$ **do**
25:       **for** $k = j + 1, \ldots, |C_i|$ **do**
26:          $d = \texttt{LPIPS}(x_{C_i[j]}, x_{C_i[k]})$       $\triangleright$ Pairwise LPIPS
27:          $T.\text{append}(d)$
28:    $D.\text{append}(\text{mean}(T))$        $\triangleright$ Average pairwise distance per cluster
29: **return** mean($D$)

---

**Algorithm 4** Calculate MC-SSIM

---

1: **Input:**
2: 1. Target Domain $Y$
3: 2. Synthesis images $I$
4: 3. Number of top scores $k$
5: **Output:** Average Top-K SSIM for each reference image
6: Initialize dictionary $D = []$        $\triangleright$ To store average SSIM per reference
7: **for** reference image $x$ in $Y$ **do**
8:    Initialize list $S = []$           $\triangleright$ To store SSIM scores
9:    **for** image $i$ in $I$ **do**
10:       $score = \texttt{SSIM}(x, i)$         $\triangleright$ Compute SSIM
11:       $S.\text{append}(score)$
12:    Sort $S$ in descending order
13:    $T = S[1 : k]$             $\triangleright$ Top-K scores
14:    **if** $T$ is not empty **then**
15:       $avg = \text{mean}(T)$
16:    **else**
17:       $avg = 0$
18:    $D[x] = avg$             $\triangleright$ Store average SSIM for $x$
19: **return** mean($D$)

---

# F ADDITIONAL EXPERIMENT RESULTS

## F.1 ADDITIONAL QUANTITATIVE EVALUATIONS

**Extended Results** To extend the results presented in Tab. 1, a more comprehensive comparison with additional methods, including TGAN Wang et al. (2018), TGAN+ADA (Karras et al., 2020), BSA Noguchi & Harada (2019), FreezeD Mo et al. (2020), EWC Li et al. (2020), CDC Ojha et al. (2021), RSSA Xiao et al. (2022), DDPM-PA Zhu et al. (2022) AdAM Zhao et al. (2022), is shown in Tab. 7.

**Diversity Quantitative Analysis** Tab. 6 presents Intra-LPIPS results. While our method not always achieve the highest scores, it is crucial to note that Intra-LPIPS has limitations in assessing true diversity. Visual artifacts can inflate this metric, potentially rewarding methods that produce diverse but low-quality outputs. Our approach prioritizes balancing diversity with fidelity to the target domain, which may not be fully captured by Intra-LPIPS alone. For a more comprehensive evaluation of generation quality, qualitative results provide additional insight (Babies: Fig. 7 and MetFaces: Fig. 8, RICK generated samples come from CRDI (Cao & Gong, 2024)).

Table 6: Comparisons Intra-LPIPS (↑) Scores between our methods and the baseline methods. Best in **bold** and the second best in **underline with bold**.

| Domains | FreezeD | RSSA | RICK | GenDA | CRDI | Ours |
|---------|---------|------|------|-------|------|------|
| Babies | 0.51 | 0.50 | **0.60** | 0.48 | 0.52 | **0.53** |
| MetFaces | 0.21 | 0.15 | 0.37 | 0.35 | **0.41** | **0.41** |

Table 7: (Extended Tab. 1) FID (↓) Scores for more baseline methods. FT represents Fine-Tuning.

| Method | Type | Babies | Sunglasses | MetFaces |
|--------|------|--------|------------|----------|
| TGAN | FT | 104.79 | 55.61 | 76.81 |
| TGAN+ADA | FT | 101.58 | 53.64 | 75.82 |
| BSA | FT | 140.34 | 76.12 | − |
| FreezeD | FT | 110.92 | 51.29 | 73.33 |
| EWC | FT | 87.41 | 59.73 | 62.67 |
| CDC | FT | 74.39 | 42.13 | 65.45 |
| RSSA | FT | 75.67 | 44.35 | 72.63 |
| DDPM-PA | FT | 48.92 | 34.75 | − |
| AdAM | FT | 48.83 | 28.03 | 51.34 |

## F.2 MORE DOMAIN ADAPTATION

To validate the performance of our method beyond the face-related domains, we performed experiments on various visual categories, including FFHQ to Otto (Yaniv et al., 2019), Church (Yu et al., 2015) to Haunted House Ojha et al. (2021), and Church to Van Gogh's house Ojha et al. (2021) adaptations. Qualitative results in Fig. 6 demonstrate consistent performance across these varied domain pairs.

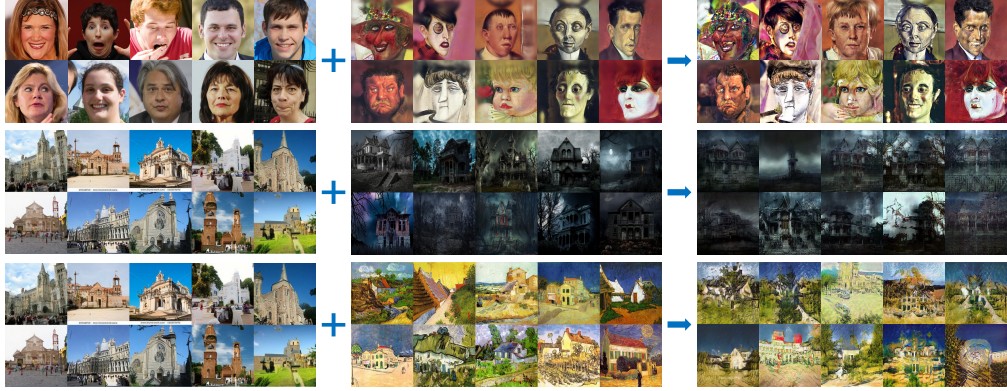

Figure 6: Adapting FFHQ → Otto (first row), Church → Haunted House (second row) and Church → Van Gogh's house (third row). First column: source domain, second column: target domain, third column: generated samples

RICK                                                    Ours

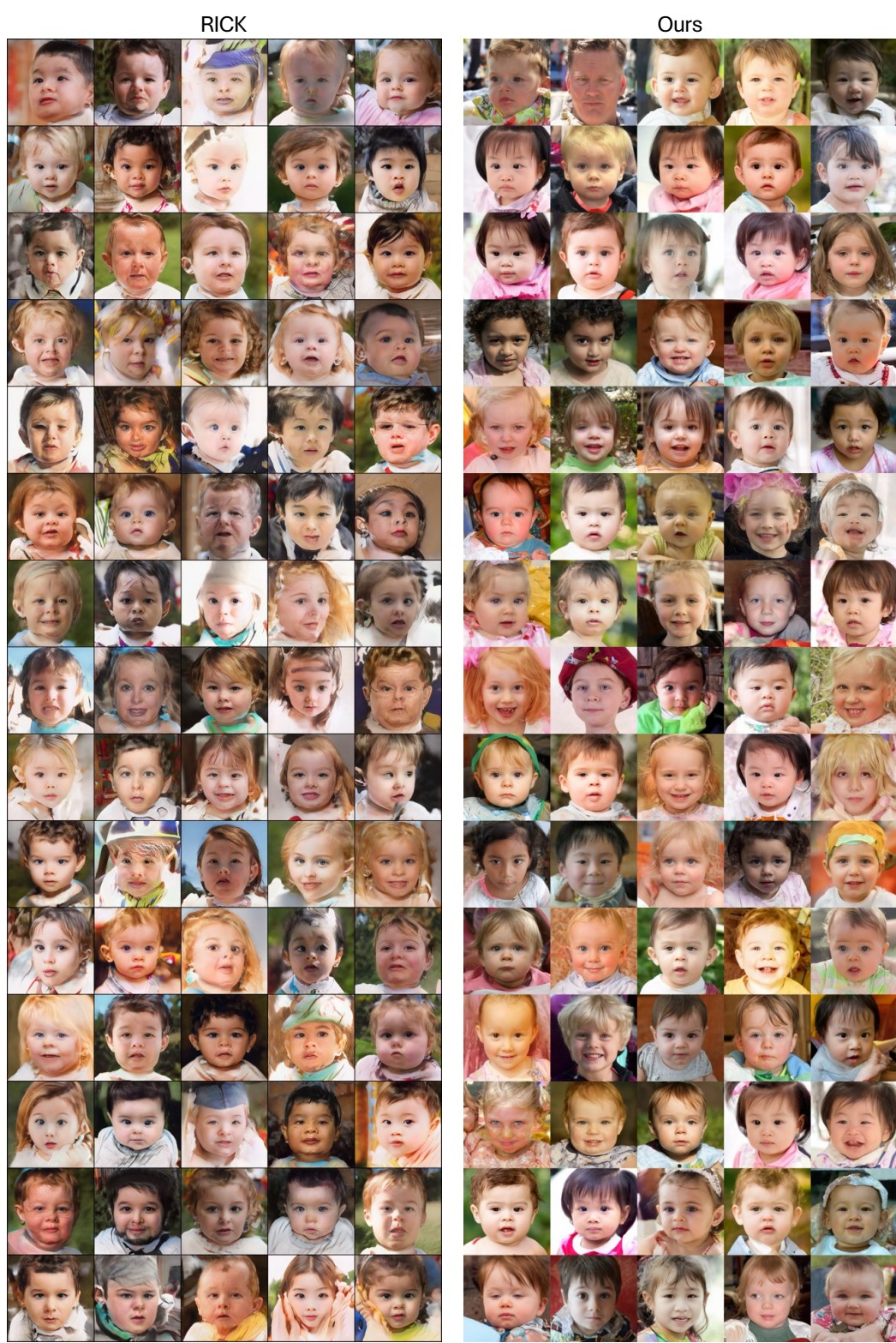

Figure 7: Qualitative comparison with RICK (state-of-the-art) on Target Domain Babies.

RICK Ours

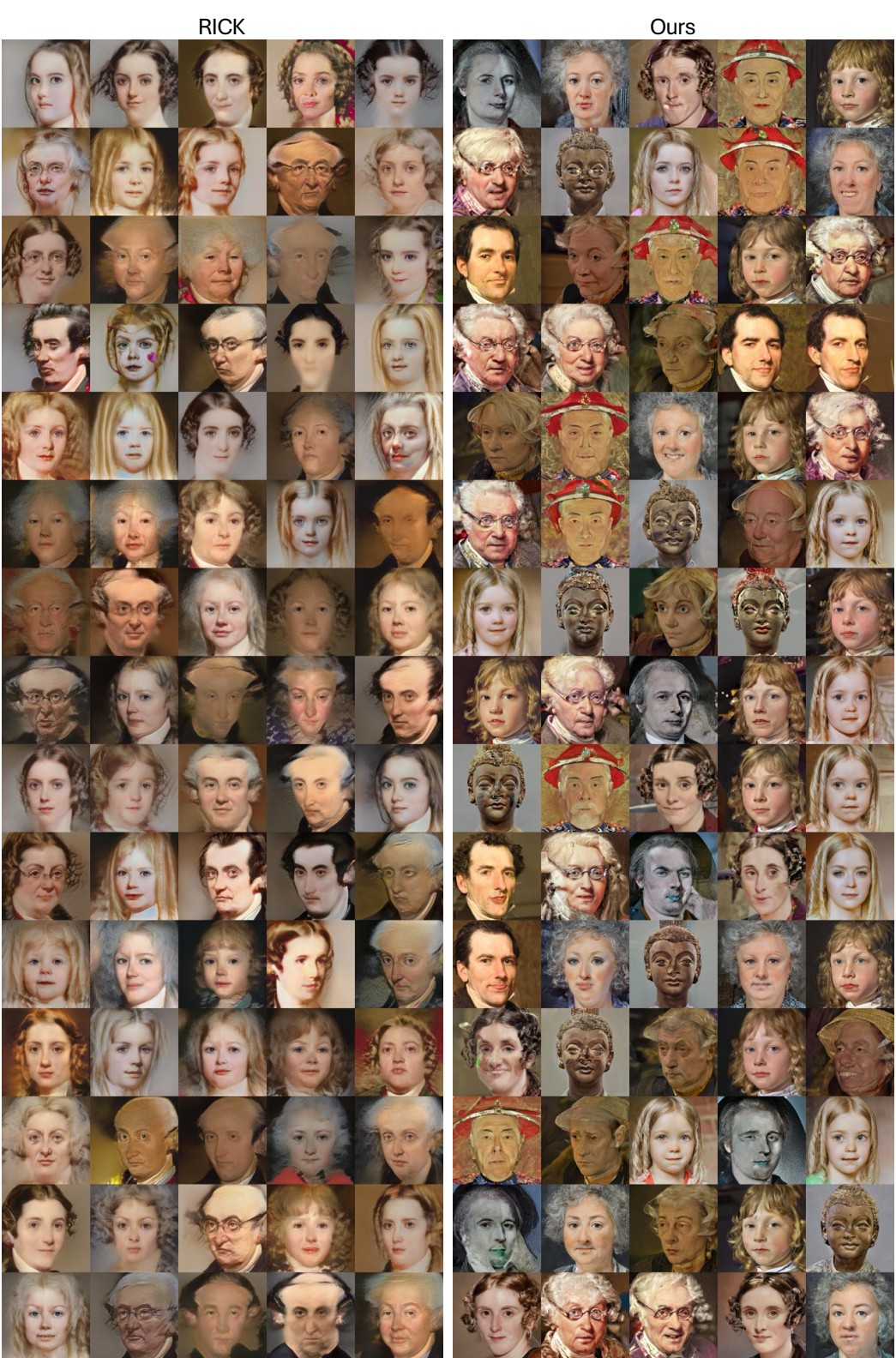

Figure 8: Qualitative comparison with RICK (state-of-the-art) on Target Domain MetFaces.

