# OpenReview forum: "EXPLORING FEW-SHOT IMAGE GENERATION WITH MINIMIZED RISK OF OVERFITTING"
_ICLR.cc/2025/Conference — ICLR 2025 Conference Withdrawn Submission_

### Official Review · Reviewer_FgyM · 2024-10-27

**Soundness:** 2
**Presentation:** 2
**Contribution:** 2
**Rating:** 5
**Confidence:** 5

**Summary:**

This paper presents Few-Shot Diffusion-regularized Representation Learning (FS-DRL), an innovative approach designed to minimize the risk of over-fitting while preserving distribution consistency in target image adaptation.

**Strengths:**

1. The proposed method is neat and seems to be effective.
2. The theoretical support is solid.

**Weaknesses:**

1. The experiments are mainly about face, which is a not very challenging task. Although a few results are provided in the supplementary, more expeirmental results about more scenes and categories should be provided.
2. There is no efficiency comparision with baselines.
3. The discussion on the related works of few-shot image generation is not comprehensive.
4. The authors only compare with three few-shot image generation baselines, which is far from enough. More recent methods should be compared.
5. There is no discussion on the failure cases.

**Questions:**

See the weakness.

---

### Official Review · Reviewer_D5zF · 2024-11-04

**Soundness:** 2
**Presentation:** 2
**Contribution:** 2
**Rating:** 5
**Confidence:** 4

**Summary:**

Focusing on the overfitting problem for few-shot image generation tasks, this paper proposes a framework for the few-shot diffusion-regularized representation learning (FS-DRL). Specifically, this framework consists of two novel parts: (1) A novel scalable Invariant Guidance Matrix (IGM) during the diffusion process, which acts as a regularizer in the feature space of the model; (2) A controllable parameter called sharing degree, which determines how many target images correspond to each IGM. Extensive experiments demonstrate that the proposed framework can effectively mitigate overfitting, enabling efficient and robust few-shot learning across diverse domains.

**Strengths:**

(+) The topic of this paper, i.e., few-shot image generation with diffusion model, is interesting and significant.
(+) This paper is easy to follow and the presentation of this paper is clear.

**Weaknesses:**

(-) The concept of overfitting in this paper seems confusing. The term "overfitting" is commonly associated with discriminative tasks rather than generative tasks. Specifically, in the training GANs with limited data, existing approaches only demonstrate the overfitting of the discriminator (D) issue, with no mention of the overfitting in the generator (G). Instead, the generator usually suffers from the gradient vanishing or instability problem. Furthermore, in diffusion models, denoising score matching is used to update the parameters of the diffusion model, which indicates that the diffusion model also suffers from the gradient issue rather than the so-called overfitting problem. The effectiveness of the proposed gradient clipping in Line 94 also demonstrates that alleviating the gradient problem is useful. Thus, only using the term “overfitting” without clearly establishing its relevance to diffusion models is inappropriate.

(-) The authors repeatedly claim that their method can analyze the degree of overfitting in diffusion models. However, concrete evidence to support this cannot be found in the paper. Which specific metric does the author use to analyze the degree of overfitting in diffusion models? One clear example, in the ADA paper, the probability p is used to indicate the overfitting degree of the discriminator. I suggest that the authors provide more supporting evidence for their claim.

(-) The authors state that their proposed method also acts as the regularizer in the diffusion model. Given that applying a regularizer to alleviate overfitting is commonly used, the novelty of this paper is not strong enough. Furthermore, I cannot find the comparison experiments between the proposed regularizer and the existing regularizer to demonstrate that the proposed method is indeed effective.

**Questions:**

Please see the weakness part.

---

### Official Review · Reviewer_dpNj · 2024-11-04

**Soundness:** 3
**Presentation:** 2
**Contribution:** 2
**Rating:** 5
**Confidence:** 4

**Summary:**

The paper addresses the challenges of deep generative models (DGMs) in few-shot image generation, particularly focusing on the issue of overfitting with extremely limited samples. It introduces Few-Shot Diffusion-Regularized Representation Learning (FS-DRL), which utilizes an Invariant Guidance Matrix (IGM) and a controllable parameter called "sharing degree" to mitigate overfitting risks.

**Strengths:**

1.	The use of the IGM as a regularizer in feature space offers a novel perspective on addressing overfitting in few-shot scenarios.
2.	The theoretical analysis of IGM enhances the interpretability of the method and provides a basis for future research.
3.	The introduction of the sharing degree parameter allows for a quantifiable balance between overfitting risk and model flexibility, improving adaptability.

**Weaknesses:**

1.	The method's reliance on both IGM and sharing degree might complicate the implementation process. Users may face challenges in tuning these additional parameters.
2.	The selection of the sharing degree is crucial, yet the paper does not provide comprehensive guidelines on how to choose this parameter effectively.
3.	Although the method shows improvements in representation learning, the computational cost associated with training and tuning, especially with higher resolutions and batch sizes, may limit its practical application in resource-constrained environments.

**Questions:**

1.	The paper would benefit from clearer guidelines or methodologies for selecting the sharing degree. Is there a way to simplify or automate the process of selecting this parameter?
2.	Since MC-SSIM is used as a metric, why were only the evaluation results for MetFaces provided?

---

### Official Review · Reviewer_vpcK · 2024-11-04

**Soundness:** 3
**Presentation:** 3
**Contribution:** 3
**Rating:** 5
**Confidence:** 5

**Summary:**

This paper proposes a new formulation for using diffusion models in few-shot image generation and introduces a novel method. Experiments show that the generated results outperform existing methods.

**Strengths:**

1. This paper presents an interesting new formulation for few-shot image generation using diffusion models, viewing Few-Shot Image Generation (FSIG) as a conditional generation problem and deriving a direct learning approach for an  Invariant Gradient Matrix (IGM) to achieve FSIG, which is innovative.
2. Experiments show that the generated results outperform existing methods.

**Weaknesses:**

1. My main concern is that the method proposed in this paper is highly sensitive to hyperparameters, specifically \gamma. From Figure 3, it can be observed that as \(\gamma\) increases, the FID first decreases and then increases, and the optimal value varies across different datasets. I suspect that even within the same domain, the optimal value may differ between datasets, which would significantly limit the applicability of this method.
2. This paper does not discuss how the sharing mechanism works when \gamma is less than the number of images. This is worth exploring.
3. When \gamma is not large enough, the model underfits, which may be related to the number of learnable parameters or the capacity of the model. This part could explore other ways to alleviate the issue of underfitting.

**Questions:**

Please refer to the "Weaknesses" section.

---

### Note · Authors · 2024-11-15

I have read and agree with the venue's withdrawal policy on behalf of myself and my co-authors.